# Sweet Syndrome Following SARS-CoV2 Vaccination

**DOI:** 10.3390/vaccines9111212

**Published:** 2021-10-20

**Authors:** Maria Efenesia Baffa, Roberto Maglie, Neri Giovannozzi, Francesca Montefusco, Stefano Senatore, Daniela Massi, Emiliano Antiga

**Affiliations:** 1Department of Health Sciences, Section of Dermatology, University of Florence, 50125 Florence, Italy; roberto.maglie@unifi.it (R.M.); francesca.montefusco@unifi.it (F.M.); stefano.senatore@unifi.it (S.S.); emiliano.antiga@unifi.it (E.A.); 2Department of Health Sciences, Section of Pathological Anatomy, University of Florence, 50139 Florence, Italy; nerigiova35@gmail.com (N.G.); daniela.massi@unifi.it (D.M.)

**Keywords:** Sweet’s syndrome, mRNA vaccines, neutrophilic dermatoses, adverse event

## Abstract

Vaccines are today considered one of the most effective means against the Sars-CoV-2 pandemic. The BNT162b2 vaccine by Pfizer/BioNTech has been massively administered throughout the globe; since its approval, a wide spectrum of cutaneous reactions has been reported. Here we report the case of a 52-year-old Caucasian male who presented with an acute febrile eruption that arose 72 h after the first dose of the BNT162b2 vaccine. The clinicopathological findings were consistent with Sweet’s syndrome. The short latency time suggested a possible role of the vaccine in triggering Sweet’s syndrome in this case.

## 1. Introduction

Since its outbreak in late 2019, coronavirus disease 19 (COVID-19) has resulted in over 4.8 million casualties [1]. Vaccines are today considered the most effective means to dampen a pandemic that has caused devastating medical, social, and economic consequences [2]. On 2 December 2020, a temporary emergency use authorization for Pfizer/BioNTech (BTN162b2) vaccine was approved in the UK; thereafter, numerous other approvals followed rapidly, leading to its massive use throughout the globe [3,4].

Since then, with the increasing number of vaccinated people, a variety of skin reactions to the BTN162b2 vaccine has been reported [5]. Here we present the case of a 52-year-old Caucasian male who presented with an acute febrile skin eruption after the first dose of the BTN162b2 vaccine.

## 2. Case

A 52-year-old man was referred to our department for an acute skin eruption that developed 72 h after the first dose of the Pfizer/BioNTech BNT162b2 mRNA vaccine.

His personal anamnesis was significant for a well-differentiated retroperitoneal liposarcoma treated in 2016 with radical surgery followed by adjuvant chemotherapy. The patient had no history of SARS-CoV-2 infection.

Physical examination revealed well demarcated, oval, tender, juicy plaques on the neck, trunk, and upper limbs. Most of them showed overwhelming vesicles and pustules, while others had a target-like appearance due to a central hemorrhagic crust (Figure 1).

Mucous membranes were not involved. The patient was febrile (constantly above 39 °C) since the onset of the skin rash despite a daily intake of paracetamol 1 g. Additional symptoms included pain and dysesthesias of the distal phalanges of the hands.

Our differential diagnosis included Sweet’s syndrome and linear IgA bullous dermatosis. Thus, a 5 mm punch biopsy was obtained from a lesion of the trunk for histopathological examination. A second skin biopsy was taken from the perilesional skin for direct immunofluorescence (DIF), alongside a peripheral blood sample for indirect immunofluorescence (IIF). Histopathology demonstrated a dense and diffuse mixed inflammatory infiltrate with a predominance of neutrophils, with subtle perivascular nuclear dust, dilatated capillaries, and prominent edema of the upper dermis (Figure 2). Either DIF or IIF gave negative results. Collectively, the clinicopathological findings were consistent with a diagnosis of Sweet’s syndrome.

As soon as he received the diagnosis, the patient proceeded to report the event to the pharmacovigilance system.

The patient received methylprednisolone (4 mg/kg/die) intravenously for three consecutive days, followed by a tapering course of oral prednisone.

The patient achieved complete resolution of the skin rash within 3 weeks of treatment.

## 3. Discussion

BNT162b2 is a nucleoside mRNA vaccine formulated in lipidic nanoparticles, which deliver the non-replicating RNA into host cells, leading to transient expression of the SARS-CoV-2 S antigen [6]. The dose schedule consists of two 30 μg doses administered 21 days apart. A phase III trial, including 43,548 participants, reported mild–moderate cutaneous reactions, mainly consisting of redness and swelling at the injection site [7].

Real-world data show a wide spectrum of post-vaccination cutaneous reactions, including local injection site reactions, urticaria, and morbilliform eruptions, pernio/chilblains, and a number of immune-mediated dermatoses [8].

To our knowledge, only three cases of Sweet’s syndrome after Covid-19 vaccines have been reported [9,10,11] and only one of them after the BNT162b2 vaccine [10] (Table 1). One of them, as in our case, had only cutaneous involvement and was related to BNT162b2; the other two also displayed extracutaneous manifestations such as arthritis, polymyositis, acute encephalitis, and myoclonus and were associated with other COVID-19 vaccines.

Sweet’s syndrome, also known as “acute febrile neutrophilic dermatosis,” is characterized by tender erythematous plaques and nodules with concomitant systemic symptoms such as fever, arthralgia, or malaise and diffuse neutrophilic infiltrate in the papillary dermis [12]. Triggers of Sweet’s syndrome include malignancies, medication, infections [13], and also vaccines [14,15,16,17,18]. Sweet’s syndrome is considered a prototype of autoinflammatory disorders due to the pathogenic role of innate immunity cytokines, including IL-1, TNF-alfa, and IL-6 [19]. In addition, a dysregulated acquired immunity also contributes to Sweet’s syndrome pathogenesis, with increased activation of Th17 and Th1 cells in the lesional skin [20]. Histopathologically, due to the presence of variable nuclear dust around post-capillary venules, a close relationship between Sweet’s syndrome and leukocytoclastic vasculitis has been postulated [21].

Vaccines have been proposed as immune-mediated diseases triggers; moreover, they usually contain adjuvants, substances that enhance the innate immune system and boost T cell activation. The mRNA contained in the novel vaccines serves both as an immunogen and adjuvant, being directly responsible for the onset of immune-mediated adverse events [22,23].

On the other hand, some authors observed that the dermatologic reactions to mRNA vaccines are often mimickers of those that occurred during SARS-CoV2 infection itself, strengthening the hypothesis that the immune response to the virus is responsible for most of its manifestations [8]. Accordingly, some cases of Sweet’s syndrome following SarS-CoV-2 infection have been reported [24,25].

In conclusion, we reported a case of Sweet’s syndrome developed shortly after the first dose of an mRNA vaccine against SARS-CoV2 infection. The short latency time in this case strongly supports the pathogenic role of the vaccine as a cause of Sweet’s syndrome in our patient. In the near future, it is expected that the use of mRNA vaccines will increase significantly. As a result, further accumulation of immune-mediated skin reactions induced by this new technology is of paramount importance.

## Figures and Tables

**Figure 1 vaccines-09-01212-f001:**
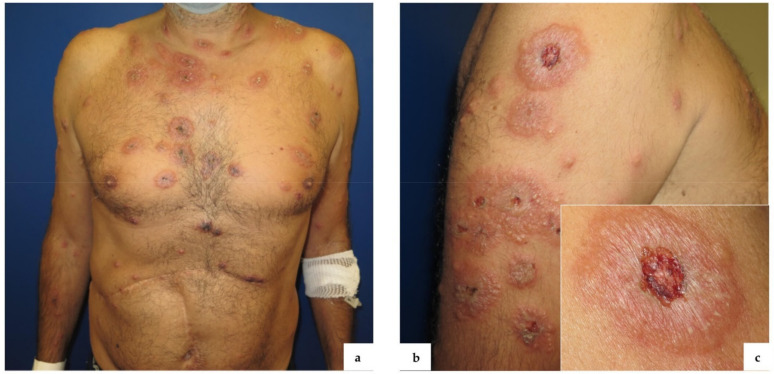
Well demarcated, targetoid lesions diffused to the trunk, neck (**a**), and upper limbs (**b**); BNT162b2 vaccine injection site (**c**).

**Figure 2 vaccines-09-01212-f002:**
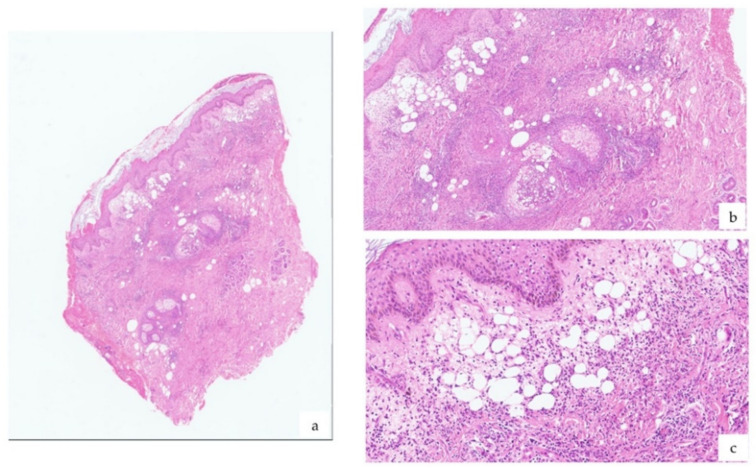
Histopathological examination of a lesion showed a dense inflammatory infiltrate and edema of the papillary dermis (**a**) (H&E); up-close views (**b**,**c**) (H&E).

**Table 1 vaccines-09-01212-t001:** Sweet’s syndrome cases reported after Covid-19 vaccination.

	Age	Sex	Administered Vaccine	Symptoms Latency	Histological Confirmation	Associated Extracutaneous Manifestations	Other Investigations
**Capassoni et al.**	37	F	ChAdOx1 nCoV-19 vaccine (Oxford-AstraZeneca)	96 h	YES	Arthritis; polymyositis	Immunofluorescence; ultrasounds; electromyography
**Darrigade et al.**	45	F	SARS-CoV-2 Pfizer-BioNTech mRNA vaccine (BNT162b2)	24 h	YES	None	Patch tests; IDR ^1^
**Torrealba-Acosta et al.**	77	M	Moderna mRNA-1273 vaccine	24 h	YES	Acute encephalitis; myoclonus	CSF ^2^ analysis

^1^ IDR, intradermoreaction; ^2^ CSF, cerebrospinal fluid.

## Data Availability

Data are available upon request to the corresponding author.

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
