# Peer review of "Sweet Syndrome Following SARS-CoV2 Vaccination"

_vaccines, 2021, doi:10.3390/vaccines9111212_

Round 1

Reviewer 1 Report

Dear Authors,

your paper entitled “Sweet’s syndrome following SARS-CoV2 vaccination” is interesting, but I believe that it needs some revisions before it can be considered for possible publication, as for my comments listed here below for the different sections of the manuscript.

Best regards,

the Reviewer.

Abstract

  1. “Vaccines are today considered one of the most effective against Sars-CoV-2 pandemic”

Something mised after effective”.

  1. In general the English language of the abstract needs professional revisions.
  2. Avoid repetitions: e.g. “presented” “presented”
  3. I would not use the verb “cause” in this case, as we don’t know causal relations and this is only a case report.

1.Introduction

  1. Line 20: unfortunately COVID-19 data become old quite fast, and now we have more that 4.7 milion deaths. Please update the data if you can. Moreover, ref. 1 weblink does not work. Perhaps you can consider also WHO: https://covid19.who.int/
  2. Lines 22-24: you refer to extra-UE approvals. Shouldn’t it be appropriate to refer also to AIFA and/or EMA approvals as your study is conducted in Europe?

2.Case

  1. Please indicate whether it was known or not if the patients had a SARS-CoV-2 infection before vaccination and when.ù
  2. Please indicate whether the adverse reaction was reported to the pharmacovigilance system.

3.Discussion

  1. Table 1 seems not following the Author guidelines in terms of graphic.
  2. Lines 73-74: it can be interested to add that, on the other hand, some cases of Sweet’s Syndrome have been reported as associated to SARS-CoV-2 infection. See e.g. https://pubmed.ncbi.nlm.nih.gov/34159397/ ; or https://pubmed.ncbi.nlm.nih.gov/32452542/
  3. Are there differences/similarities between your case and the other described in scientific literature? E.g. considering extracutaneous manifestations did you observe something?

References

  1. n° 1, n° 6 link do not work
  2. n° 2, 3 and 4 link do not report the news you refer to.
  3. All the refs do not follow the style of the journal: please revise.

Author Response

Dear Reviewer 1,

Thank you for your revision and for the interest in our work. Please find below our point-by-point reply:

  1. We add "means" after "effective" (line 8).
  2. We revised English in the abstract (line 8-15).
  3. We tried to avoid repetitions in the abstract (line 11).
  4. We changed the last sentence of the abstract in order to avoid the use of the verb "cause" (line 13-14).
  5. We updated COVID data as well as the link (ref. 1).
  6. Following your suggestion, we included reference to European approval (ref. 3)
  7. As suggested, in the case description, we reported that "The patient had no history of SARS-CoV-2 infection" (line 34-35).
  8. The patients reported the adverse event to the pharmacovigilance system. We included this information within the text (line 57-58).
  9. We changed Table1 in order to make it more adherent to the “Authors guidelines” of the journal (line 82-83).
  10. As suggested, we added that some cases of Sweet's syndrome occurred following SARS-CoV-2 infection. Consequently, we included ref. 23 and 24 in the reference list.

11. As suggested, we included a sentence highlighting the main differences and similarities between our case and those reported in the literature so far (line 101-107).

  1. We provided working links for the references you indicated (line 122, line 134).
  2. We checked the links and changed as suggested (line 123-130).
  3. We revised the reference style.

Reviewer 2 Report

  1. Minor English corrections: line 35 must read "oval", not "ovalar"; line of figure 1 must read "limbs", not "libs"; line 65/6 must in my view read "deliver", not "delivery".
  2. What was the exact diagnosis of retropertitoneal malignancy: liposarcoma, lymphoma,...?
  3. For skin lesions term "juicy" lesions is best adequate, please, and also nicely visable.
  4. Please, omit or relativate the view on histology once more. The process is accentuated around areas of postcapillary venules with neutrophils and nuclear dust, and, admittedly no larger amounts of fibrin. Capillary regions show edeme in papillary dermis, but less cellularity. In my view you should not say there was no evidence of leukocytoclastic vasculitis. Please, also see (and when necessary for text cite) literature on Sweet`s syndrome by Ratzinger G et al, also including the thoughts about the vasculitic wheel.

Author Response

Dear Reviewer 2,

We would like to thank you for the careful revision of our manuscript. Please find below our point-by-point reply:

(1) We checked the language mistakes you flagged (line 36, line 42, line 68-69).

(2) As requested, we added the exact diagnosis that was that of "well differentiated retroperitoneal liposarcoma" (line 33-34).

(3) As suggested, we included the term "juicy" in the description of the lesions (line 36)

(4) We deleted the sentence related to absence of leukocytoclastic vasculitis. In addition, we added the suggested reference including the comments on vasculitic wheel (line 95-57).